# Significant inefficiency in running community health systems: The case of health posts in Southwest Ethiopia

**Kiddus Yitbarek**[1]*, **Gelila Abraham**[1], **Melkamu Berhane**[2], **Sarah Hurlburt**[3], **Carlyn Mann**[3], **Ayinengida Adamu**[4], **Gebeyehu Tsega**[4], **Mirkuzie Woldie**[1,3]

**1** Department of Health Policy and Management, Institute of Health, Jimma University, Jimma, Ethiopia, **2** Department of Pediatrics and Child Health, Institute of Health, Jimma University, Jimma, Ethiopia, **3** Department of Global Health and Population, Harvard T.H. Chan School of Public Health, Boston, MA, United States of America, **4** Department of Public Health, Bahirdar University, Bahirdar, Ethiopia

* kiddus.yitbarek@yahoo.com

## Abstract

### Background

Although much has been documented about the performance of the health extension program, there is a lack of information on how efficiently the program is running. Furthermore, the rising cost of health services and the absence of competition among publicly owned health facilities demands strong follow up of efficiency. Therefore, this study aimed to assess the technical efficiency of the health posts and determinants in Southwestern Ethiopia.

### Methods and materials

We used data for one Ethiopian fiscal year (from July 2016 to June 2017) to estimate the technical efficiency of health posts. A total of 66 health posts were included in the analysis. We employed a two-stage data envelopment analysis to estimate technical efficiency. At the first stage, technical efficiency scores were calculated using data envelopment analysis program version 2.1. Predictors of technical efficiency were then identified at the second stage using Tobit regression, with STATA version 14.

### Results

The findings revealed that 21.2% were technically efficient with a mean technical efficiency score of 0.6 (± 0.3), indicating that health posts could increase their service volume by 36% with no change made to the inputs they received. On the other hand, health posts had an average scale efficiency score of 0.8 (± 0.2) implying that the facilities have the potential to increase service volume by 16% with the existing resources. The regression model has indicated average waiting time for service has negatively affected technical efficiency.

**Data Availability Statement:** All relevant data are within the paper and its Supporting Information files.

**Funding:** The funder of this project is Jimma University. Although the funders support the study they had no role in study design, data collection and analysis, decision to publish, or preparation of the manuscript.

**Competing interests:** The authors have declared that no competing interests exist.

## Conclusion

More than three-quarters of health posts were found inefficient. The technical efficiency score of more than one-third of the health posts is even less than 50%. Community mobilization to enhance the uptake of health services at the health posts coupled with a possible reallocation of resources in less efficient health posts is a possible approach to improve the efficiency of the program.

## Introduction

Globally the healthcare system still falls short of providing good-quality, accessible, and integrated care. The problems are more serious in low-income countries [1–3]. These countries are however suffering from resource shortage to give services at an acceptable quality. Even though there is increase in health care spending, according to a recent projection, even the growth of total and per capita public health spending will not show a significant increase for the future [4–6]. This makes the journey of these countries towards the global goal of universal health coverage (UHC) as part of the Sustainable Development Goals (SGD) difficult [7, 8].

This warrants a call for an increased interest in access and utilization of primary health care (PHC) services. Many countries around the world use community health intervention as an effective and cost-saving strategy to raise access to PHC [9]. Especially in a resource-limited setting, this approach has resulted in extensive improvements to health care access and equity [10, 11]. It has a long term cost implication to the health system while preventing the adulthood and diseases related to aging [12]. Community health systems are usually used to deliver preventive, promotive, and basic curative services. However, the system and overall administration of the services are quite different among various countries. In some countries, the services are being delivered on voluntary bases, and in some other countries in a well-structured system and paid health cadres [13, 14]. What so ever the approach community health service has a great benefit to prevent various health problems. It has also an impact on the cost of care during the adulthood and old age periods [15].

Ethiopia was among the first countries to implement PHC services after the Alma Atta declaration in 1978 [16]. Services are administered through the primary health care unit (PHCU), which consists of health posts (HPs), health centers, and primary hospitals [17]. Health posts are at the first point of contact and the most accessible health service delivery units to the community in rural areas [18]. In 2004, the Ethiopian Federal Ministry of Health (EFMOH) introduced the community health extension program (HEP). This program initiated the establishment of HPs in every *kebele* (the smallest administrative division in Ethiopia) with the placement of two or more trained female health extension workers (HEWs) to staff the HPs. The HEWs are health workers that receive a salary and at times housing is provided for them, unlike community-based health worker programs in other countries which traditionally have been on a volunteer basis and more recently are provided non-financial incentives [19]. This program is believed to have improved access to essential preventive health services, promote the health of communities, and contributed to the reduction in under-5 child mortality [20]. HEWs are trained to provide sixteen health packages, categorized into four major groups, to the community in their catchment area [18, 21, 22].

As in other developing countries, Ethiopia is already faced with financial constraints given its current health service delivery scope and even more so as it moves towards universal health coverage [17, 23, 24]. This lack of resources is coupled with different system-wide problems

leading to significant unmet healthcare need in terms of service availability, access, and quality [25–27]. A significant part of these problems could be addressed by being considerate of the cost of healthcare and using the available resources more efficiently. The shortage of relevant studies on the efficiency of the health care system in Ethiopia hinders health system policy and decision-makers to allocate resources wisely [28]. On top of that there is a high demand for efficiency studies both in the implementation and academic world to ensure the transferability of knowledge [15].

Technical efficiency (TE) is in health care organizations attained as the volume of inputs (like human resource, supplies, and drugs) minimized for a given level of output or when the volume of output (like the number of patients treated, the number of deliveries attended) increase for a given amount of input [28]. TE is one of the major performance measures of the health sector. It can be measured with different approaches using inputs and outputs variables of the health sector. The most used and well-known approach of measurement in the data envelopment analysis (DEA). It is a mathematical linear programming model that builds efficiency frontiers and discriminate among efficient and inefficient decision-making units (DMUs) [29, 30]. The DEA tool is used to benchmarking the performance of health facilities in terms of health resources and outputs. In this approach, TE can be measured in constant returns to scale models (CRS) and variable returns to scale (VRS). In the VRS model, the inefficiency can be because of pure technical inefficiency or scale inefficiency [31, 32].

The public sector is the major health service provider in many low-income countries, including Ethiopia [33, 34]. Since public health service providers do not compete with each other, which could create an incentive to improve both quality and efficiency, there should be a mechanism to monitor and improve the efficiency of public facilities. Although the HEP has contributed to promising improvements in many health indicators in the country, its performance ought to be advanced to ensure achievements in national and international goals. As one of the performance measures, there is a lack of information on the efficiency of the HPs. Therefore, we aimed to assess the TE of the HPs and its determinants in Southwestern Ethiopia.

## Methods

### Study setting and design

We conducted a cross-sectional study of 72 HPs located in eight districts of Jimma Zone, in the Oromia Region, Southwest Ethiopia, from 19 March to 28 April 2018. There are a total of twenty districts and one town administration in the zone. These districts are further divided into 548 kebeles (the smallest administrative unit in Ethiopia), among them 515 (93.98%) are rural. The projected total population of the zone, based on the 2007 Ethiopian census, is about 3.56 million in 2018. There are 5 primary hospitals, 115 health centers and 520 HPs situated in the zone. HPs are at the first points of contact in the Ethiopian three-tiered health system. It is where most of the HEP packages are delivered and managed. Each HP is staffed with at least two HEWs and serves an average of 5000 population. Five satellite HPs operate under a PHCU that consists of a health center and a primary hospital. The overall service delivery of HEWs is supervised by the health center episodically [18, 35].

### Variables

Input variables were; the number of HEWs and non-salary recurrent expenses (expenses for supplies, drugs, and vaccine). The non-salary expenditure data were collected in Ethiopian birr and changed into the US dollar with the August 2018 exchange rate of Ethiopian Birr (ETB) 27.3755 to one US$ [36].

As an output variable, we have considered the number of health education sessions, antenatal care (ANC), family planning (FP) services, child immunization, treated diarrhea cases, treated malaria cases, number of households (HH) visits, and number of referral.

Variables for Tobit regression: we have considered the TE score of each HP as a dependent variable. The independent variables were (1) service years of the HP: is a continuous variable referring the number of years since the establishment of the HP; (2) size of the catchment population: is a continuous variable referring the number of population living near the HP who are under the catchment area of the HP; (3) availability of health facility around: availability of another health facility within the kebele where the HP is located and (4) average waiting time for service: the mean time elapsed in the HP to get service.

We have visited all the study HPs to collect data on inputs, outputs, and additional predictor variables from their records, with resource inventory and interview with HEWs. We then visited health centers that supervise the respective HP and got back to the district health office and the finance and economic cooperation office (specifically for financial information) to verify the information obtained from each HP.

## Data collection

The health facilities included in the study were identified using the Tools for Assessing the Operationality of District Health Systems. It suggests a sample of 40% of the districts if the number of districts is between 20 and 29. Furthermore, at least three health facilities from each district are suggested to be included [37]. Hence, eight out of 20 districts (40%) in Jimma Zone were included in the study. Within these districts, 24 primary health care units (PHCUs) were sampled and three HPs were selected from each PHCU. This totaled to 72 HPs that were included in the study.

Tools for data collection were developed after consulting a guide for the Ethiopian HEP and other relevant literatures [38–40]. Afterward, the tools were translated into Afaan Oromo, the local working language, and translated back into English by an independent translator to check for semantic equivalence and consistency. The tools were designed to collect data on input and output items and organizational and environmental factors thought to be predictors of TE. We have conducted a resource inventory, document review, and an interview with head HEW in each HP. Data was collected for the Ethiopian fiscal year 2009 (July 2016 to June 2017). Seven trained data collectors and three supervisors have undertaken the overall data collection fieldwork.

## Data analysis

From the 72 HPs included in the study 66 HPs with complete information were identified for this analysis. Descriptive statistics were performed using Stata version 14. A two-stage DEA was performed. DEA program (DEAP) (version 2.1), a DEA computer program developed by Tim Coelli [31], was used to estimate the TE scores—the first stage analysis. We have specified our DEA model as described in a former article by Bobo FT. et al. [41]. We have employed the VRS model of TE in this study. This model was preferred assuming not all HPs are considered to be operating at an optimal scale. This is referred to as scale inefficiency and takes two forms–Decreasing Returns to Scale (DRS) and Increasing Returns to Scale (IRS) [39, 42]. We also used an output-oriented DEA model because HEWs in HPs have more control over the volume of outputs rather than resources allocated to the HP [43, 44].

**DEA conceptual framework.**   Data envelopment analysis is a linear programming model that measures the relative efficiency of DMUs (in our case health posts) using multiple inputs

and multiple outputs [45, 46].

$$Technical\ Efficiency = \frac{Weighted\ sum\ of\ outputs}{Weighted\ sum\ of\ inputs}$$

*Orientation.* In this study, we employed an output TE measure, since HEWs can better influence outputs than inputs. HEWs can raise the number of health service users by their efforts including promotions, creating better connections in the community, and with better access to the home visit. With this study, we have the objective of recommending HEWs the necessary actions they have to do to raise the number of service users. Our measure answers the question: "By how much can the volume of output quantities be proportionally raised without changing the input quantities used?" The choice of the approach is recommended to be based on which side of the orientation (input or outputs) the decision-makers in the health facility have more control over [42, 47].

*Model specification.* We have both constant and variable returns to scale models to measure the relative efficiency of HPs. In the CRS mode, we have an assumption that in the production process the optimum mix of inputs and outputs is independent of the scale of production. The DEA CRT scores of TE can be disintegrated into two components, one due to "pure" technical inefficiency and the other due to scale inefficiency [31, 48].

If the health facilities are not operating at an optimal scale, the TE measure will be mixed with scale efficiency. Hence, to separate the pure TE and scale efficiency scores VRS model is considered. The VRS is the extension of the CRS model equation after imposing a convexity constraint on it. This means that the data are enveloped more closely than the CRS model. The model then compares the efficiency same size DMUs. Thus, the relative efficiency score of health facilities can be obtained by using the following equation as given by Cooper A. et al [48, 49]:

$$Efficiency = Max \sum_r U_r y_{rjo} + U_0$$

Subject to

$$\sum_r U_r y_{rj} - \sum_r V_i X_{ij} + U_0 \leq 0 \quad ; j = 1, \ldots, n$$

$$\sum_i V_i X_{ijo} = 1$$

$$U_r, V_i \geq 0$$

Where:

Yrj = the amount of output r produced by health facility j,

Xij = the amount of input I used by health facility j,

Ur = the weight given to output r, (r = 1... t and t is the number of outputs),

Vi = the weight given to input I, (I = 1... m and m is the number of inputs),

j0 = the health facility under assessment

We employed the VRT model in this study because the interest is on the extent to which the scale of operations affects productivity or when not all units of analysis are considered to be operating at an optimal scale. This is as a result of scale inefficiency and takes two forms–

Increasing Returns to Scale (IRS) and Decreasing Returns to Scale (DRS). IRS implies that a health facility is too small for the volume of activities that it operates. To function at the most productive scale size, a health facility exhibiting DRS should expand its scale of operation to become scale efficient. In contrast, a health facility with DRS is too large for its scale of operation. If a DMU is exhibiting DRS, it should scale down its scale of operation [42, 50].

**Tobit regression.** The second stage of the analysis consisted of running a Tobit regression model to identify the independent predictors of HPs' TE scores. Considering that the efficiency scores fall between 0 and 1, and several scores tend to concentrate on these boundary values (censored at 1). For this reason, Tobit regression is a good estimator of explanatory variables [51, 52]. Statistical significance was declared for a significance level of 0.05, and the 95% confidence interval of coefficients [53].

We used a Tobit model to identify determinants of TE because the TE scores are censored at 1. We say variable Y is censored when X is observed for all cases, however, the true value of Y falls at some restricted range of observations. The observations spread above or below the restricted range. There are two conditions here (1) If $Y \geq m$ for all Y, then Y is left-censored or censored from below. (2) When $Y \leq m$ for all Y, then Y is right-censored or censored from above [52]. In our case for instance the efficiency scores (Y) of observations fall below or equal to 1. In this case the ordinary least square (OLS) measure could be biased. We consider the standard linear equation, but we see the normality only from one side. Tobit regression is a modified likelihood function that reflects the uneven sampling probability for each observation based on whether the latent dependent variable falls below or above some threshold [51].

$$y = x\beta + u$$

The distribution is normal at $(0, \sigma 2)$, but we only see $w = \min(y,c)$ if censored from the above, or $w = \max(y,c)$ if censored from below [49].

### Ethical considerations

The study was approved by the institutional review board (IRB) of the Institute of Health, Jimma University. We have obtained informed consent from leaders of each HP before data collection.

## Results

### Characteristics of health posts

In this study, a total of 72 HPs were considered and we got complete information for 66 (91.7%). All the studied HPs are located in the rural kebeles of Jimma zone and led by female HEWs. About 36 (54.6%) of the HPs are the only health facilities in their respective kebeles, while the remaining 30 (45.5%) are located in a kebele where health center is available. The HPs in this study served an average population of 5814 with a mean service years of nine. The mean waiting time for service in the studied HPs was 21.4 minutes. Averagely about two HEWs serve in each of the HP (Table 1).

**Table 1. Characteristics of health posts in Southwest Ethiopia, 2018.**

| Variable | Mean | SD | Min | Max |
|---|---|---|---|---|
| Service years of the HP | 9.7 | 2.5 | 5 | 18 |
| Size of the Catchment population | 5814.0 | 2387.6 | 1042 | 12283 |
| Average waiting time for service (minutes) | 21.4 | 10.5 | 2 | 45 |
| Number of HEWs | 2.2 | 0.8 | 1 | 4 |

## Technical efficiency of health posts

Fourteen (21.2%) HPs were found to be technically efficient as a result of both pure technical and scale efficiency. Among all HPs, 22 (33.3%) were found to have pure TE and 19 (28.8%) were scale efficient. The mean efficiency score for all HPs was 0.6 ± 0.3. The mean pure TE score for all HPs was 0.8 ± 0.2 and the mean scale efficiency score was 0.8 ± 0.2. When we looked at the returns to scale, 19 (28.8%) HPs were operating in a constant returns to scale, while 5 (7.6%) were operating in increasing returns to scale and 42 (63.6%) were operating in decreasing returns to scale [S1 Table] (Table 2).

## Potential output increase of health posts

Almost three quarters of the HPs were inefficient, with a potential for input reduction or output increase. Keeping the health inputs as they are, averagely a single inefficient HP could increase its health education sessions by 20, number of ANC by 72; FP services by 304; diarrheal case treatments by 34; household visits by 467; malaria case treatments by 7; child vaccination services by 141 and referral services by 12 [S2 and S3 Tables] (Table 2).

In total, there were 143 HEWs working in the included 66 HPs. This is slightly above the average of 2 HEWs per HP. HPs were found to have spent US\$ 28,760.17 on drugs, supplies and vaccine. All of these resources were used to provide health services in their catchment. In total, there were 2,106 health education sessions conducted by 143 health extension workers throughout 12 months of the assessment. There were also 8,482 ANC services, 40,603 FP services, 14,203 child immunization services, 4,829 diarrhea treatments, 529 malaria cases treatments, 58,067 household visits, and 3,075 referral services rendered in the HPs. We found a significant mean difference among outputs of efficient and inefficient HPs' (Table 3).

## Factors associated with technical efficiency of health posts

As the information in Table 4 indicates, we found a negative statistical association between average waiting time for service and TE [β = -0.0089, 95% CI, -0.0166, -0.0013]. We did not find any significant statistical association between any other predictor variables and TE scores.

## Discussion

As one of the most popular approaches in low and middle-income countries, community-based health service delivery has resulted in remarkable improvements in access and equity in using primary health services. Consequently, many countries tend to invest a significant

**Table 2. Summary of technical efficiency of health posts and potential output increase in Southwestern Ethiopia, 2016/17.**

| | CRS_TE | VRS_TE | SE | Potential output increase | | | | | | | |
|---|---|---|---|---|---|---|---|---|---|---|---|
| | | | | Health Education Sessions | ANC | FP service | Diarrhea treated | HH visit | Malaria treated | Children Immunization | Referral services |
| **Sum** | | | | 1335 | 4725 | 20096 | 2244 | 30821 | 446 | 9321 | 12 |
| **Mean** | 0.6 | 0.8 | 0.8 | 20 | 72 | 304 | 34 | 467 | 7 | 141 | 20 |
| **SD** | 0.3 | 0.2 | 0.2 | 18 | 75 | 322 | 35 | 508 | 12 | 146 | 784 |
| **Min** | 0.2 | 0.2 | 0.5 | 0 | 0 | 0 | 0 | 0 | 0 | 0 | 0 |
| **Max** | 1.0 | 1.0 | 1.0 | 53 | 230 | 1167 | 140 | 1680 | 47 | 646 | 111 |

CRS_TE: Constant returns to scale technical efficiency (overall technical efficiency).

VRS_TE: Variable returns to scale technical efficiency (pure technical efficiency).

SE: Scale efficiency.

**Table 3. Description of inputs used and outputs produced in health posts of Southwestern Ethiopia, 2016/17.**

| Variables | Efficient HPs (n = 14) | | | Inefficient HPs (n = 52) | | | P—value |
|---|---|---|---|---|---|---|---|
| | Mean | SD | Sum | Mean | SD | Sum | |
| **Input** | | | | | | | |
| HEW | 2.1 | 1.0 | 30 | 2.2 | 0.8 | 113 | 0.9 |
| Non salary expense* (US$) | 161.7 | 177.2 | 2264.2 | 509.5 | 1188.5 | 26495.9 | 0.3 |
| **Output** | | | | | | | |
| Health Education Sessions | 38 | 29.6 | 531 | 30 | 29.1 | 1575 | 0.4 |
| ANC | 145 | 102.8 | 2028 | 124 | 93.1 | 6454 | 0.5 |
| FP service | 828 | 670.4 | 11595 | 558 | 317.7 | 29008 | 0.03 |
| Children Immunization | 379 | 321.6 | 5310 | 171 | 107.6 | 8893 | < 0.01 |
| Diarrhea treated | 125 | 137.8 | 1753 | 59 | 59.5 | 3076 | 0.01 |
| Malaria treated | 11 | 16.7 | 149 | 7 | 18.3 | 380 | 0.5 |
| Household visit | 1221 | 808.2 | 17096 | 778 | 708.0 | 40971 | 0.05 |
| Referral | 142 | 315.8 | 1989 | 21 | 78.9 | 1086 | 0.01 |

*expenses for drugs, supplies and vaccine.

amount of their health care expenses to expand the service [54–56]. To sustain the benefits from community health services, their performance has to be followed up and improved [13]. Their performance really makes a significant difference in the health indicators of a country [57]. Various countries use different dimensions to measure their health system's performance. The most used are quality, efficiency, equity, and effectiveness [54]. As one of the performance measures, we have assessed the TE of community health services in the Southwestern part of Ethiopia.

Our study indicates that nearly three-quarters of sampled HPs in southwest Ethiopia were operating below the estimated capacity of best performing HPs, due to inefficiencies in the operation and scale of production. The TE scores of the sampled HPs range from 0.2 to 1, indicating an 82% difference in resource utilization. On average, HPs could increase their service volume by 36% without any need for additional resources. Five HPs (7.6%) would be required to increase their scale of operation, while 42 (63.6%) HPs could scale back their operations because they are too large for the volumes of activities conducted, to be efficient.

The use of DEA has become a more widely used method to measure TE and to help inform policy reforms to improve health system performance. A similar study to ours was conducted in northern Ethiopia found that 75% of HPs were operating inefficiently [39]. In a study of

**Table 4. Tobit regression result of technical efficiency of health posts, Southwest Ethiopia, 2016/17.** Tobit regression. Number of obs = 66. LR chi2(4) = 9.21. Prob > chi2 = 0.06. Log likelihood = -26.22.

| Variables | Coef. | Std. Err. | t | P>t | [95% CI] | |
|---|---|---|---|---|---|---|
| Service years of HP | -0.01 | 0.02 | -0.8300 | 0.41 | -0.05 | 0.02 |
| Size of the catchment population | -3E-05 | 2E-05 | -1.78 | 0.08 | -6.3E-05 | 3.58E-06 |
| Availability of health facility around the HP<br>Yes<br>No | -0.07 | 0.09 | -0.84 | 0.40 | -0.24 | 0.10 |
| Average waiting time * | -0.01 | 0.004 | -2.34 | 0.02 | -0.02 | -0.001 |
| _cons | 1.21 | 0.19 | 6.31 | 0 | 0.83 | 1.60 |
| /sigma | 0.30 | 0.03 | | | 0.24 | 0.37 |

*significant at p–value < 0.05.

health centers in Southwestern Ethiopian, 50% of health centers were found to be technically inefficient [41]. Results from Kenya and Sierra Leone revealed that 56% and 79% of primary health centers were inefficient, respectively [58, 59]. Two studies conducted for Ghana's primary health care system found that 69% and 78% of inefficient health centers [49, 60]. A study of HPs in Guatemala found efficient HPs dropped from 53% to 29% between 2008 and 2009 [40].

As compared to the above findings, the inefficiency in our study seems higher. Unlike our study, all have assessed the TE at the primary health center level except the one conducted in northern Ethiopia [39]. Although the difference is small, the efficiency of HPs in northern Ethiopia is still better than the southwestern, according to the findings. The difference could be because; the study conducted in northern Ethiopia was between 2007 and 2008, a few years from the introduction of HEP in Ethiopia. As pieces of evidence indicate, the follow-up and support to the program during those times was strong, and HEWs were well motivated to implement the packages they are supposed to execute. In recent years, the HEP is being criticized for its low productivity and efficiency [61, 62].

Health posts in our study have a potential to increase their service delivery with the current number of health extension workers and non-salary expenditures, consisting of 63.4% more health education sessions, 55.7% more ANC, 49.5% more FP services, 46.5% more diarrhea treatments, 53.1% more household visits, 84.2% more malaria case treatments, 65.6% more child immunization services, and 25.5% more referral services.

Furthermore, comparing the service outputs of efficient and inefficient HPs, we found significant disparities. In most instances, there is a fixed and relatively similar distribution of resources along with HPs in Ethiopia in general and in Oromia regional state in particular [63, 64]. Therefore, practically the efficiency driver is the demand for health services rather than what happens on the supply side. HEWs and the systems' effort to increase the service volume can make a significant contribution to improving the efficiency of the HEP.

The Tobit regression analysis has found that average waiting time for service in HPs significantly affects TE. A one-minute increase in the average waiting time resulted in a 0.9% decrease in the TE score. A long waiting time was found to minimize the volume of health services used in a certain HP, compromising efficiency as a result. The size of the catchment population was found to positively affect TE of primary health centers in Ethiopia in a study by Bobo et al [41]. While in our study the variable has no significant association with TE of HPs. Other studies have found that the number of clinical and non-clinical staff, shorter lifespan of the facility, and additional incentive packages for employees were determinants of TE [41, 61]. While it is not significantly associated our study pointed out that, longer lifespan of HPs has negatively related to TE.

The readers of this study should consider the following limitations while reading this article. First, health-seeking and service utilization have a strong impact on TE. This study did not examine behavioral, social, and cultural factors that possibly have a strong influence on the outputs of the health system. Second, DEA is a comparative analysis, rates that deviate from the best practice as inefficient. Some may have used their resources to improve quality; and also topographic barriers in some part of the study area may have affected HEWs to address services to the wider proportion of the population. Finally, capital costs like building, equipment, and vehicles were not considered, because of lack and low quality of data.

## Conclusion

This study found a high level of inefficiency in providing preventive and promotive health services in the HPs of Southwest Ethiopia. More than one-third of the HPs are operating below

50% of their actual capacity. Redistribution of resources in inefficient HPs would increase TE. Decision-makers responsible for HP resource allocation should revise their allocation mechanisms to account for variations in facility efficiency—increasing the resources to efficient facilities and decreasing the resources allocated to inefficient facilities. Another approach to improve the efficiency of inefficient facilities, applied especially by HEWs, would be to strengthen efforts to increase service volume through community mobilization to enhance demand for services by the communities.

## Supporting information

**S1 Table. Technical efficiency of health posts, Southwest Ethiopia, 2018.**
(DOCX)

**S2 Table. Summary of targets and potential increase in outputs of health posts, Southwest Ethiopia, 2018.**
(DOCX)

**S3 Table. Potential output increase in inefficient health posts, Southwest Ethiopia, 2018.**
(DOCX)

**S1 Data.**
(DTA)

## Acknowledgments

We would like to thank data collectors, supervisors and respondents of the study.

## Author Contributions

**Conceptualization:** Kiddus Yitbarek, Gelila Abraham, Ayinengida Adamu, Gebeyehu Tsega, Mirkuzie Woldie.

**Data curation:** Kiddus Yitbarek, Melkamu Berhane, Mirkuzie Woldie.

**Formal analysis:** Kiddus Yitbarek.

**Funding acquisition:** Kiddus Yitbarek, Mirkuzie Woldie.

**Investigation:** Kiddus Yitbarek, Gelila Abraham, Melkamu Berhane, Ayinengida Adamu, Gebeyehu Tsega, Mirkuzie Woldie.

**Methodology:** Kiddus Yitbarek, Sarah Hurlburt, Carlyn Mann, Mirkuzie Woldie.

**Project administration:** Kiddus Yitbarek, Gelila Abraham, Melkamu Berhane, Mirkuzie Woldie.

**Software:** Kiddus Yitbarek, Sarah Hurlburt, Carlyn Mann.

**Supervision:** Kiddus Yitbarek, Ayinengida Adamu.

**Writing – original draft:** Kiddus Yitbarek.

**Writing – review & editing:** Gelila Abraham, Melkamu Berhane, Sarah Hurlburt, Carlyn Mann, Ayinengida Adamu, Gebeyehu Tsega, Mirkuzie Woldie.

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
