## [Decision Letter · Decision Letter 0]

1 Oct 2020

PONE-D-19-28271

Significant inefficiency in running community health systems: the case of the Health Extension Program in Southwest Ethiopia

PLOS ONE

Dear Dr. Yitbarek,

Thank you for submitting your manuscript to PLOS ONE. After careful consideration, we feel that it has merit but does not fully meet PLOS ONE’s publication criteria as it currently stands. Therefore, we invite you to submit a revised version of the manuscript that addresses the points raised during the review process.

The manuscript has been evaluated by three reviewers, their comments are available below.

The reviewers have raised a number of concerns, regarding the reporting of methodological aspects of the study such as insufficient explanation of the statistical analysis used. In addition, they request a careful consideration of the terminology used to ensure that this is sufficiently defined. Please also note that your manuscript requires copyediting.

Please note that reviewer #2 has also suggested the inclusion of additional references but we do not require you to include these in your next revision.

Could you please carefully revise the manuscript to address all comments raised.

We look forward to receiving your revised manuscript.

Kind regards,

Sara Fuentes Perez, PhD

Staff Editor

PLOS ONE

Journal Requirements:

2. Please refer to any post-hoc corrections to correct for multiple comparisons during your statistical analyses. If these were not performed please justify the reasons. Please refer to our statistical reporting guidelines for assistance (https://journals.plos.org/plosone/s/submission-guidelines.#loc-statistical-reporting).

"We would like to thank Jimma University for funding this study. Moreover, our gratitude goes to data collectors, supervisors and respondents."

Reviewers' comments:

Reviewer's Responses to Questions

**Comments to the Author**

1. Is the manuscript technically sound, and do the data support the conclusions?

Reviewer #1: Partly

Reviewer #2: Yes

Reviewer #3: Partly

2. Has the statistical analysis been performed appropriately and rigorously? 

Reviewer #1: I Don't Know

Reviewer #2: Yes

Reviewer #3: No

3. Have the authors made all data underlying the findings in their manuscript fully available?

Reviewer #1: Yes

Reviewer #2: Yes

Reviewer #3: No

4. Is the manuscript presented in an intelligible fashion and written in standard English?

Reviewer #1: No

Reviewer #2: Yes

Reviewer #3: No

5. Review Comments to the Author

Reviewer #1: General comments

This paper, if it is going to be accepted for publication, will need some major revisions. There are as follows:

1. A clear definition of “technical efficiency” and how it is measured. After reading the paper, I had no clear sense of what this term actually referred to. Why is this important, and what practical steps could be taken to improve technical efficiency?

2. Although the title of the paper is on the technical efficiency of the Health Extension Program, the text focuses on health posts rather than the HEWs themselves. But since the measures of technical efficiency focus on the HEWs and their work outside of the health post as well as within it (including household visits and services provided individually by HEWs in the community), the narrative should as well.

Specific major comments

1. We need a definition of “Data Development Analysis”.

2. The Tobit regression methodology needs explanation for the uninformed reader (which will be 99% of the readers of this article”.

3. Need to explain Tools for Assessing the Operationality of District Health Systems (p. 7, line 114-5).

4. Need explanation of Data Envelopment Analysis (p. 8, line 133).

Specific minor comments

1. P. 2 line 31: “severely” is not an appropriate word here.

2. P. 4 line 56: Ref 13 is the wrong ref.

3. P. 5 line 62: Ref 14 is outdated. Here is a suggestion for an up-to-date reference: Perry H. Health for the People: National Community Health Programs from Afghanistan to Zimbabwe. 2020. https://pdf.usaid.gov/pdf_docs/PA00WKKN.pdf.

4. P. 6, line 100: Should define the date at which that exchange rate was measured.

5. P. 8, line 137: You use the term TE here but not consistently for technical efficiency. There should be consistent use throughout the paper.

6. P. 10, line 154: Need to define what you mean by “near a health center”.

7. P. 10, line 144: Are you referring to how long the health post has been in operation or to how long the HEWs have been working there?

8. P. 10, line 160: There is no need for 2 decimal places anywhere in the paper.

9. P. 10, line 161: Your bring in the term “scale efficiency” without defining it.

10. P. 11, line 166: You bring in the terms increasing and decreasing returns to scale without defining them.

11. P. 12, line 180: Are you referring to the average spent by each health post?

12. P. 14, lines 209-211: This sentence is unclear.

13. P. 15, line 220: The word “sever” is inappropriate and also misspelled.

14. P. 15, lines 238-242: Not clear how this was computed.

15. P. 15, lines 246-248: Good point. But, there is a ord missing after “significant”.

16. P. 15, line 251: Should this be “decrease” rather than “increase?

Reviewer #2: This is a decent contribution on inefficiency in running community health systems in the example of Health Extension Program in Ethiopia.

It fills certain knowledge gap and is worthy of publishing.

Yet the evidence base should be signficantly expanded.

OECD academic sources dominate alongside with few national ones.

Much more LMICs and EMerging Markets documented evidence should be added to increase diversity and reliability of the claims in the text.

Thus I warmly recommend introduction of some of the following sources listed beneath:

Jakovljevic, M., Timofeyev, Y., Ranabhat, C. et al. Real GDP growth rates and healthcare spending – comparison between the G7 and the EM7 countries. Global Health 16, 64 (2020). https://doi.org/10.1186/s12992-020-00590-3

Jakovljevic, M., Matter-Walstra, K., Sugahara, T. et al. Cost-effectiveness and resource allocation (CERA) 18 years of evolution: maturity of adulthood and promise beyond tomorrow. Cost Eff Resour Alloc 18, 15 (2020). https://doi.org/10.1186/s12962-020-00210-2

Jakovljevic, M., Potapchik, E., Popovich, L., Barik, D., & Getzen, T. E. (2017). Evolving health expenditure landscape of the BRICS nations and projections to 2025. Health economics, 26(7), 844-852.

Jakovljevic, M., Jakab, M., Gerdtham, U., McDaid, D., Ogura, S., Varavikova, E., ... & Getzen, T. E. (2019). Comparative financing analysis and political economy of noncommunicable diseases. Journal of medical economics, 22(8), 722-727.

Rancic, N., & Jakovljevic, M. M. (2016). Long term health spending alongside population aging in N-11 emerging nations. East Eur Bus Econ J, 2(1), 2-26.

Jakovljevic, M., & Getzen, T. E. (2016). Growth of global health spending share in low and middle income countries. Frontiers in pharmacology, 7, 21. https://www.frontiersin.org/articles/10.3389/fphar.2016.00021/full

Dieleman, J. L., Campbell, M., Chapin, A., Eldrenkamp, E., Fan, V. Y., Haakenstad, A., ... & Reynolds, A. (2017). Future and potential spending on health 2015–40: development assistance for health, and government, prepaid private, and out-of-pocket health spending in 184 countries. The Lancet, 389(10083), 2005-2030.

Jakovljevic, M. B. (2014). The key role of the leading emerging BRIC markets in the future of global health care. Serbian Journal of Experimental and Clinical Research, 15(3), 139-143.

Jakovljevic, M. B. (2015). BRIC’s growing share of global health spending and their diverging pathways. Frontiers in public health, 3, 135.https://www.frontiersin.org/articles/10.3389/fpubh.2015.00135/full

Jakovljevic, M., Groot, W., & Souliotis, K. (2016). Health care financing and affordability in the emerging global markets. Frontiers in public health, 4, 2. https://www.frontiersin.org/articles/10.3389/fpubh.2016.00002/full

Conditional to adopting at least several of these remarks, I am willing to review the revised manuscript assuming its maturity for publishing.

Reviewer #3: General comment: The paper requires language polishing because it is hard to read.

For instance: ...and changed

100 in to US dollar ...

Statistical analysis:

Data Envelopment Analysis (DEA) was performed - Add a citation for this.

used to estimate

135 the technical efficiency scores - Describe these scores in detail.

Tobit regression model - Add a citation and a discussion of the model.

Statistical significance was declared with p-values less than 0.05, and the 95% confidence

147 interval of coefficients. - What you mean is:

Statistical significance was declared for a significance level of 0.05, and the 95% confidence

147 interval of coefficients.

Add citation for this:

https://www.mdpi.com/2504-4990/1/3/54

Table 4: From the text, the covariates used for the regression analysis are unclear. Provide this information explicitely in the table and the main text.

6. PLOS authors have the option to publish the peer review history of their article (what does this mean?). If published, this will include your full peer review and any attached files.

Reviewer #1: **Yes: **Henry B. Perry

Reviewer #2: No

Reviewer #3: No

---

## [Author Response · Author response to Decision Letter 0]

4 Oct 2020

Dear Editors,

Thank you very much for giving me the chance to revise our manuscript. I have critically read all the comments raised by the editor and reviewers, and revised the manuscript accordingly. Moreover, we have provided responses to editor’s and reviewers comments in a point by point fassion.

Journal Requirements:

Response: Thank you for the reminder. We have kept the guidelines for the revised copy. 

2. Please refer to any post-hoc corrections to correct for multiple comparisons during your statistical analyses. If these were not performed please justify the reasons. Please refer to our statistical reporting guidelines for assistance (https://journals.plos.org/plosone/s/submission-guidelines.#loc-statistical-reporting).

Response: Thank you. We were not requested to perform any corrections to the statistical analysis by the reviewers and the editor. 

"We would like to thank Jimma University for funding this study. Moreover, our gratitude goes to data collectors, supervisors and respondents."

Response: Thank you very much for the suggestion. The funder of this project is Jimma University. Although the funders support the study they had no role in study design, data collection and analysis, decision to publish, or preparation of the manuscript. We have put this in the online submission system. The acknowledgement was corrected in the revised copy.

Response: Thanks, corrected in the revised copy. 

Response: Thank you very much. The captions are now included at the final page of the main manuscript document. 

Reviewers' comments:

Reviewer's Responses to Questions

 

Comments to the Author

1. Is the manuscript technically sound, and do the data support the conclusions?

Reviewer #1: Partly

Reviewer #2: Yes

Reviewer #3: Partly

2. Has the statistical analysis been performed appropriately and rigorously?

Reviewer #1: I Don't Know

Reviewer #2: Yes

Reviewer #3: No

3. Have the authors made all data underlying the findings in their manuscript fully available?

Reviewer #1: Yes

Reviewer #2: Yes

Reviewer #3: No

4. Is the manuscript presented in an intelligible fashion and written in standard English?

Reviewer #1: No

Reviewer #2: Yes

Reviewer #3: No

5. Review Comments to the Author

Reviewer #1: General comments

Dear reviewer, 

Thank you very much for raising very important issues for the improvement of our work. That is really helpful. We have revised the manuscript based on your valuable suggestions. Bellow I have given responses for all the raised questions in a point by point fashion. 

This paper, if it is going to be accepted for publication, will need some major revisions. There are as follows:

1. A clear definition of “technical efficiency” and how it is measured. After reading the paper, I had no clear sense of what this term actually referred to. Why is this important, and what practical steps could be taken to improve technical efficiency?

Response: Thank you for the relevant suggestion. We now included description about technical efficiency in the introduction section of the revised manuscript. 

2. Although the title of the paper is on the technical efficiency of the Health Extension Program, the text focuses on health posts rather than the HEWs themselves. But since the measures of technical efficiency focus on the HEWs and their work outside of the health post as well as within it (including household visits and services provided individually by HEWs in the community), the narrative should as well.

Response: you are right the paper is on technical efficiency of health extension program. And we have used health posts as decision making units. 

Health extension workers (HEWs) are community health workers who give service to the community in different approaches including outreach services, home-to-home services and based in the health post. A health post is staffed with averagely 2 – 4 HEWs whose primary role is implementing the health extension program. There are also resources relevant for health extension program in the health posts. 

Therefore, we have taken a health post as a decision making unit (DMU) (unit of analysis in this case) instead of individual HEW, because the health extension program cannot function without considering HEWs and other relevant resources.

Specific major comments

1. We need a definition of “Data Development Analysis”.

Response: thank you for the insightful suggestion. We now included the definition in the introduction briefly and in the method of analysis sub-section in detail in the revise manuscript.

2. The Tobit regression methodology needs explanation for the uninformed reader (which will be 99% of the readers of this article”.

Response: Thank you for the suggestion. We have written explanation in the method of analysis section of the revised manuscript. 

3. Need to explain Tools for Assessing the Operationality of District Health Systems (p. 7, line 114-5).

Response: Tools for Assessing the Operationality of District Health Systems is a document that states how we should take sample size in a district and health facility based studies. The suggestions were given by WHO. It has suggestions of number of health facilities and districts to be sampled based on the total number of districts and health service organizations (population) in the study area. The sampling of this study was performed based on the guideline.

4. Need explanation of Data Envelopment Analysis (p. 8, line 133).

Response: Thank you. Detailed explanation was given under the method of analysis sub-section of the revised manuscript.

Specific minor comments

1. P. 2 line 31: “severely” is not an appropriate word here.

Response: Thank you. Corrected in the revised copy.

2. P. 4 line 56: Ref 13 is the wrong ref.

Response: Thank you, corrected in the revised copy.

3. P. 5 line 62: Ref 14 is outdated. Here is a suggestion for an up-to-date reference: Perry H. Health for the People: National Community Health Programs from Afghanistan to Zimbabwe. 2020. https://pdf.usaid.gov/pdf_docs/PA00WKKN.pdf.

Response: Thank you very much. We have replaced the reference in the revised manuscript.

4. P. 6, line 100: Should define the date at which that exchange rate was measured.

Response: Thank you for the suggestion. We have used the August 2018 exchange rate. Included in the revised copy.

5. P. 8, line 137: You use the term TE here but not consistently for technical efficiency. There should be consistent use throughout the paper.

Response: Thank you for the suggestion. Corrected in the entire document except in the abstract and titles of the revised manuscript. 

6. P. 10, line 154: Need to define what you mean by “near a health center”.

Response: Thank you, it is to mean “in a kebele where health center is available”. Described in the revised copy.

7. P. 10, line 144: Are you referring to how long the health post has been in operation or to how long the HEWs have been working there?

Response: in the specified line, we have described about the nature of the efficiency scores. The efficiency scores take values between 0 and 1. To come up with these scores we considered inputs like number of health extension workers working in the specific health post and non-salary recurrent expenditures. 

8. P. 10, line 160: There is no need for 2 decimal places anywhere in the paper.

Response: thank you. We have changed to one decimal place except the monetary values and some very low p-values and confidence intervals, in the revised copy.

9. P. 10, line 161: Your bring in the term “scale efficiency” without defining it.

Response: now it is defined well under DEA conceptual framework section of the revised manuscript. 

10. P. 11, line 166: You bring in the terms increasing and decreasing returns to scale without defining them.

Response: now it is defined well under DEA conceptual framework section of the revised manuscript. 

11. P. 12, line 180: Are you referring to the average spent by each health post?

Response: At the point of the study, health posts are expected to be staffed with an average of 2 HEWs. However, practically there were some health posts that have more than 2. Therefore, 143 HEWs for 66 health posts is slightly above two. 

HEWs and other resources were seen as a separate input variables in the study. 

12. P. 14, lines 209-211: This sentence is unclear.

Response: We have added sentences before the mentioned sentence. Now it seems better for understanding. 

13. P. 15, line 220: The word “sever” is inappropriate and also misspelled.

Response: Thank you, corrected in the revised copy.

14. P. 15, lines 238-242: Not clear how this was computed.

Response: Thank you for this important question. Potential output increase was described in table 2. The description in the mentioned page and line is the proportion of potential output increase taking the actual outputs as a denominator. 

15. P. 15, lines 246-248: Good point. But, there is a word missing after “significant”.

Response: Thank you, corrected in the revised copy.

16. P. 15, line 251: Should this be “decrease” rather than “increase?

Response: Thank you, corrected in the revised copy.

 

Reviewer #2

Dear reviewer,

Thank you very much for the suggestions and reference materials you provided to me. They are really helpful. 

Response: However, most of them are macro-economic analyses, despite the micro-economic nature of our paper. Our paper studied the technical efficiency (a process measure) of Ethiopian community health service under the primary health care. We therefore, added other relevant references that complies with our manuscript. 

Reviewer #2: This is a decent contribution on inefficiency in running community health systems in the example of Health Extension Program in Ethiopia.

It fills certain knowledge gap and is worthy of publishing. Yet the evidence base should be signficantly expanded.

OECD academic sources dominate alongside with few national ones. Much more LMICs and EMerging Markets documented evidence should be added to increase diversity and reliability of the claims in the text.

Thus I warmly recommend introduction of some of the following sources listed beneath:

Jakovljevic, M., Timofeyev, Y., Ranabhat, C. et al. Real GDP growth rates and healthcare spending – comparison between the G7 and the EM7 countries. Global Health 16, 64 (2020). https://doi.org/10.1186/s12992-020-00590-3

Jakovljevic, M., Matter-Walstra, K., Sugahara, T. et al. Cost-effectiveness and resource allocation (CERA) 18 years of evolution: maturity of adulthood and promise beyond tomorrow. Cost Eff Resour Alloc 18, 15 (2020). https://doi.org/10.1186/s12962-020-00210-2

Jakovljevic, M., Potapchik, E., Popovich, L., Barik, D., & Getzen, T. E. (2017). Evolving health expenditure landscape of the BRICS nations and projections to 2025. Health economics, 26(7), 844-852.

Jakovljevic, M., Jakab, M., Gerdtham, U., McDaid, D., Ogura, S., Varavikova, E., ... & Getzen, T. E. (2019). Comparative financing analysis and political economy of noncommunicable diseases. Journal of medical economics, 22(8), 722-727.

Rancic, N., & Jakovljevic, M. M. (2016). Long term health spending alongside population aging in N-11 emerging nations. East Eur Bus Econ J, 2(1), 2-26.

Jakovljevic, M., & Getzen, T. E. (2016). Growth of global health spending share in low and middle income countries. Frontiers in pharmacology, 7, 21. https://www.frontiersin.org/articles/10.3389/fphar.2016.00021/full

Dieleman, J. L., Campbell, M., Chapin, A., Eldrenkamp, E., Fan, V. Y., Haakenstad, A., ... & Reynolds, A. (2017). Future and potential spending on health 2015–40: development assistance for health, and government, prepaid private, and out-of-pocket health spending in 184 countries. The Lancet, 389(10083), 2005-2030.

Jakovljevic, M. B. (2014). The key role of the leading emerging BRIC markets in the future of global health care. Serbian Journal of Experimental and Clinical Research, 15(3), 139-143.

Jakovljevic, M. B. (2015). BRIC’s growing share of global health spending and their diverging pathways. Frontiers in public health, 3, 135.https://www.frontiersin.org/articles/10.3389/fpubh.2015.00135/full

Jakovljevic, M., Groot, W., & Souliotis, K. (2016). Health care financing and affordability in the emerging global markets. Frontiers in public health, 4, 2. https://www.frontiersin.org/articles/10.3389/fpubh.2016.00002/full

Conditional to adopting at least several of these remarks, I am willing to review the revised manuscript assuming its maturity for publishing.

 

Reviewer #3

Dear reviewer,

Thank you very much for you time reviewing our manuscript and your valuable inputs. We made changes to the manuscript based on your suggestions. Bellow we have given responses to all the raised questions.

Reviewer #3: General comment: The paper requires language polishing because it is hard to read.

Response: Thank you very much for the suggestion. We did a copy edit to polish the grammatical errors and write-up issues.

For instance: ...and changed

100 in to US dollar ...

Response: Thank you. Corrected in the revised copy.

Statistical analysis:

Data Envelopment Analysis (DEA) was performed - Add a citation for this.

Response: Thank you very much. We put citation and further explanation for the model. 

used to estimate

135 the technical efficiency scores - Describe these scores in detail.

Response: Thank you very much. We have given sufficient explanation about DEA model and TE scores under the “DEA conceptual framework” sub-title.

Tobit regression model - Add a citation and a discussion of the model.

Response: Thank you for the insightful suggestion. We have put a sufficient explanation about the Tobit model with the relevant citation in the revised copy. 

Statistical significance was declared with p-values less than 0.05, and the 95% confidence

147 interval of coefficients. - What you mean is:

Statistical significance was declared for a significance level of 0.05, and the 95% confidence

147 interval of coefficients.

Add citation for this:

https://www.mdpi.com/2504-4990/1/3/54

Response: Thank you very much for the suggestion. We made changes and added the citation in the revised copy according to the suggestion in the revised manuscript.

Table 4: From the text, the covariates used for the regression analysis are unclear. Provide this information explicitely in the table and the main text.

Response: Thank you for the suggestion. We put further description of the variables in the “Methods” section, under “Variables” sub heading in the revised copy.

---

## [Decision Letter · Decision Letter 1]

30 Nov 2020

PONE-D-19-28271R1

Significant inefficiency in running community health systems: the case of the Health Extension Program in Southwest Ethiopia

PLOS ONE

Dear Dr. Yitbarek,

Thank you for submitting your manuscript to PLOS ONE. After careful consideration, we feel that it has merit but does not fully meet PLOS ONE’s publication criteria as it currently stands. Therefore, we invite you to submit a revised version of the manuscript that addresses the points raised during the review process.

We look forward to receiving your revised manuscript.

Kind regards,

Abhijit P Pakhare, M.D.

Academic Editor

PLOS ONE

Additional Editor Comments (if provided):

Manuscript have been improved from its earlier version. However, some points still needs to be clarified.

How was the waiting time determined?

Whether distance from nearest PHCU was explored and evaluated? Distance is an important accessibility barrier which may affect service uptake.

Service output has constraints due to population size. For example, number of ANCs can’t be increased than existing birth rate. So, output has to be scaled or redefined considering population, crude birth rate and expressed in terms of percentage of ANC examination vis a vis expected examinations as per guidelines. Thus, number of ANCs will be driven by their population size. Same logic applies to other health services as well like FP services, diarrheal case, malaria case treatments, child vaccination services, referral services and also household visits.

Thus, catchment population seems to be an effect modifier and not only confounder which can be adjusted by multivariate analysis. Consider sub-group analysis by stratifying according to population size

Also, comments of the reviewers needs to be addressed.

Reviewers' comments:

Reviewer's Responses to Questions

**Comments to the Author**

1. If the authors have adequately addressed your comments raised in a previous round of review and you feel that this manuscript is now acceptable for publication, you may indicate that here to bypass the “Comments to the Author” section, enter your conflict of interest statement in the “Confidential to Editor” section, and submit your "Accept" recommendation.

Reviewer #1: All comments have been addressed

Reviewer #2: (No Response)

Reviewer #4: (No Response)

2. Is the manuscript technically sound, and do the data support the conclusions?

Reviewer #1: Yes

Reviewer #2: Partly

Reviewer #4: Yes

3. Has the statistical analysis been performed appropriately and rigorously? 

Reviewer #1: I Don't Know

Reviewer #2: N/A

Reviewer #4: Yes

4. Have the authors made all data underlying the findings in their manuscript fully available?

Reviewer #1: Yes

Reviewer #2: Yes

Reviewer #4: Yes

5. Is the manuscript presented in an intelligible fashion and written in standard English?

Reviewer #1: Yes

Reviewer #2: Yes

Reviewer #4: Yes

6. Review Comments to the Author

Reviewer #1: I have reviewed the responses to the comments I made on review of the initial submission. I support the acceptance and publication of this paper in its revised form.

Reviewer #2: Dear Authors,

Please read again carefully my previous Review.

Neither a single ONE out of my recommendations have been adopted.

Thus I firmly believe this Manuscript revision is not mature for publishing in its current appearance.

Sincerely

Reviewer #4: Overall:

This is an important contribution. After the revisions, this version of the paper presents a clear and consistent analysis.

However, I have 2 major comments which the authors need to address before publication:

1. As indicated in the earlier round of review by Reviewer 1, there is a disconnect between the title and the contents of the manuscript. The title suggests that the assessment is far broader than what the content actually addresses. The title should be revised to reflect this for 2 reasons:

(a) It is about the inefficiency in the HE program, which is captured through an evaluation of the the health posts as far as I can see.

(b) It does not get into any analysis, qualitative or quantitative, of inefficiencies of community health programmes in general.

2. For interpreting the associations that the econometric exercise throws up, it is very important to be upfront about two things: (a) Caution about establishing causality among variables by making clear the assumptions in using the health posts related data to reflect the HE program, the latter being at a programmatic level as opposed to the former. (b) The limitations or caveats in interpreting the findings as an assessment of the health extension programme and, in general to a community health programme, have to be adequately acknowledged and mentioned in both the introduction and the concluding part of the paper.

7. PLOS authors have the option to publish the peer review history of their article (what does this mean?). If published, this will include your full peer review and any attached files.

Reviewer #1: **Yes: **Henry B Perry

Reviewer #2: No

Reviewer #4: No

---

## [Author Response · Author response to Decision Letter 1]

6 Jan 2021

PONE-D-19-28271R1

Significant inefficiency in running community health systems: the case of the Health Extension Program in Southwest Ethiopia

PLOS ONE

Dear Dr. Yitbarek,

Thank you for submitting your manuscript to PLOS ONE. After careful consideration, we feel that it has merit but does not fully meet PLOS ONE’s publication criteria as it currently stands. Therefore, we invite you to submit a revised version of the manuscript that addresses the points raised during the review process.

We look forward to receiving your revised manuscript.

Kind regards,

Abhijit P Pakhare, M.D.

Academic Editor

PLOS ONE

 

Additional Editor Comments (if provided):

Manuscript have been improved from its earlier version. However, some points still needs to be clarified.

How was the waiting time determined?

Response: Dear Prof. Pakhare, thank you for the concern. Health facilities including primary health centers and health posts have a record of waiting time for service. They finally calculate the average waiting time for service in their respective health institution and report it to the primary health care unit head with other reportable indicators. Therefore, we took the average waiting time of each of these health posts as appeared on their official report to the primary health care unit (PHCU).

Whether distance from nearest PHCU was explored and evaluated? Distance is an important accessibility barrier which may affect service uptake.

Response: Thank you for the question. We actually did not measure the distance of the nearest health facility or PHCU. We rather seen the availability of health facility near the health post. This was measured by checking the availability of another health facility within the Kebele (Kebele is the smallest administrative unit in Ethiopia) where the HP is located. 

Service output has constraints due to population size. For example, number of ANCs can’t be increased than existing birth rate. So, output has to be scaled or redefined considering population, crude birth rate and expressed in terms of percentage of ANC examination vis a vis expected examinations as per guidelines. Thus, number of ANCs will be driven by their population size. Same logic applies to other health services as well like FP services, diarrheal case, malaria case treatments, child vaccination services, referral services and also household visits. Thus, catchment population seems to be an effect modifier and not only confounder which can be adjusted by multivariate analysis. Consider sub-group analysis by stratifying according to population size

Response: Sure, the size of the catchment population can be an effect modifier to the service outputs. However, we did not use the service outputs as an outcome variable. We used this variable to measure efficiency of health posts. The analysis we employed to assess the technical efficiency of health posts was a non-parametric test, data envelopment analysis (DEA). We used variables like number of health extension workers as input and variables like number of ANC as output to run the DEA. The outcome variable for this analysis is technical efficiency. The size of the catchment population might be an effect modifier to the output rather than the outcome variable (technical efficiency). 

 

Comments to the Author

1. If the authors have adequately addressed your comments raised in a previous round of review and you feel that this manuscript is now acceptable for publication, you may indicate that here to bypass the “Comments to the Author” section, enter your conflict of interest statement in the “Confidential to Editor” section, and submit your "Accept" recommendation.

Reviewer #1: All comments have been addressed

Reviewer #2: (No Response)

Reviewer #4: (No Response)

2. Is the manuscript technically sound, and do the data support the conclusions?

Reviewer #1: Yes

Reviewer #2: Partly

Reviewer #4: Yes

3. Has the statistical analysis been performed appropriately and rigorously?

Reviewer #1: I Don't Know

Reviewer #2: N/A

Reviewer #4: Yes

4. Have the authors made all data underlying the findings in their manuscript fully available?

Reviewer #1: Yes

Reviewer #2: Yes

Reviewer #4: Yes

5. Is the manuscript presented in an intelligible fashion and written in standard English?

Reviewer #1: Yes

Reviewer #2: Yes

Reviewer #4: Yes

6. Review Comments to the Author

 

Reviewer #1: I have reviewed the responses to the comments I made on review of the initial submission. I support the acceptance and publication of this paper in its revised form.

Thank you very much

 

Reviewer #2: Dear Authors,

Please read again carefully my previous Review. Neither a single ONE out of my recommendations have been adopted. Thus I firmly believe this Manuscript revision is not mature for publishing in its current appearance.

Response: Thank you very much for raising these important issues and providing essential literature. Now we have answered your concerns in the revised copy.

 

Reviewer #4: Overall:

This is an important contribution. After the revisions, this version of the paper presents a clear and consistent analysis.

However, I have 2 major comments which the authors need to address before publication:

1. As indicated in the earlier round of review by Reviewer 1, there is a disconnect between the title and the contents of the manuscript. The title suggests that the assessment is far broader than what the content actually addresses. The title should be revised to reflect this for 2 reasons:

(a) It is about the inefficiency in the HE program, which is captured through an evaluation of the the health posts as far as I can see.

(b) It does not get into any analysis, qualitative or quantitative, of inefficiencies of community health programmes in general.

Response: thank you for this relevant concern. Now we modified the title in a way that indicates the whole paper.

2. For interpreting the associations that the econometric exercise throws up, it is very important to be upfront about two things: (a) Caution about establishing causality among variables by making clear the assumptions in using the health posts related data to reflect the HE program, the latter being at a programmatic level as opposed to the former. (b) The limitations or caveats in interpreting the findings as an assessment of the health extension programme and, in general to a community health programme, have to be adequately acknowledged and mentioned in both the introduction and the concluding part of the paper.

Response: You are right. In this study we did analysis on health posts and just thought a proxy for health extension program. Based on reviewers’ suggestions and discussion among ourselves, we decided to make the entire focus on health posts. However we have shown the implications of the findings in relation to community health services.

---

## [Decision Letter · Decision Letter 2]

22 Jan 2021

Significant inefficiency in running community health systems: the case of Health Posts in Southwest Ethiopia

PONE-D-19-28271R2

Dear Dr. Yitbarek,

We’re pleased to inform you that your manuscript has been judged scientifically suitable for publication and will be formally accepted for publication once it meets all outstanding technical requirements.

Kind regards,

Abhijit P Pakhare, M.D.

Academic Editor

PLOS ONE

Additional Editor Comments (optional):

All comments have been responded.

Reviewers' comments:

Reviewer's Responses to Questions

**Comments to the Author**

1. If the authors have adequately addressed your comments raised in a previous round of review and you feel that this manuscript is now acceptable for publication, you may indicate that here to bypass the “Comments to the Author” section, enter your conflict of interest statement in the “Confidential to Editor” section, and submit your "Accept" recommendation.

Reviewer #4: All comments have been addressed

2. Is the manuscript technically sound, and do the data support the conclusions?

Reviewer #4: Yes

3. Has the statistical analysis been performed appropriately and rigorously? 

Reviewer #4: Yes

4. Have the authors made all data underlying the findings in their manuscript fully available?

Reviewer #4: Yes

5. Is the manuscript presented in an intelligible fashion and written in standard English?

Reviewer #4: Yes

6. Review Comments to the Author

Reviewer #4: The paper has adequately explained the comments that were raised.

As of now, the paper makes a value addition to the subject matter, especially for Ethiopia.

7. PLOS authors have the option to publish the peer review history of their article (what does this mean?). If published, this will include your full peer review and any attached files.

Reviewer #4: No

---

## [Editor Report · Acceptance letter]

1 Feb 2021

PONE-D-19-28271R2 

Significant inefficiency in running community health systems: the case of Health Posts in Southwest Ethiopia 

Dear Dr. Yitbarek:

I'm pleased to inform you that your manuscript has been deemed suitable for publication in PLOS ONE. Congratulations! Your manuscript is now with our production department. 

Kind regards, 

on behalf of

Dr. Abhijit P Pakhare 

Academic Editor

PLOS ONE